# Peptide AEDL and Glutathione Stimulates Root Development *Nicotiana tabacum*

**DOI:** 10.3390/ijms26010289

**Published:** 2024-12-31

**Authors:** Neonila Vasilievna Kononenko, Larisa Ivanovna Fedoreyeva

**Affiliations:** All-Russia Research Institute of Agricultural Biotechnology, Timiryazevskaya 42, 127550 Moscow, Russia; nilava@mail.ru

**Keywords:** GSH, peptide AEDL, ROS, stem cells

## Abstract

Reactive oxygen species (ROS) are essential molecules involved in intercellular communication, signal transduction, and metabolic processes. Abiotic stresses cause the accumulation of excess ROS in plant cells. The issue of regulating the antioxidant protection of plants using natural and synthetic compounds with antioxidant activity still remains one of the most important and relevant areas of fundamental and applied research. Glutathione (GSH) plays an important role in the stress resistance and redox homeostasis of plant cells and effectively protects the cell from the stress-induced generation of ROS. An increase in the GSH content in plant cells can contribute to an increase in plant resistance to various types of stressors. We have shown that growing *Nicotiana tabacum* in the presence of tetrapeptide AEDL (AlaGluAspLeu) contributes to an increase in the GSH content by 3.24 times. At the same time, the tobacco plant was more developed, especially its root system. A scheme of the mechanism behind the regulation of the redox balance in the stem cell niche and the participation of the AEDL and GSH peptides in the regulation of the fate of stem cells was proposed.

## 1. Introduction

Oxygen is an essential element required for normal plant development [1]. As a result of oxidation–reduction reactions occurring in various compartments of plant cells, ROS are formed due to the incomplete or partial reduction of oxygen molecules [2]. Excessive production of ROS, if not removed, can lead to so-called “oxidative stress”. The term “oxidative stress” can be used when the levels and types of oxidants in a cell or organelle, on average, significantly exceed the content associated with normal homeostatic function for each compartment or cell, since each has its own oxidation–reduction balance [3,4]. ROS accumulation most often occurs near electron transport chains. In plants, the accumulation of excess ROS occurs near the thylakoid membranes of chloroplasts, in which the process of photosynthesis takes place [5] and the internal membranes of mitochondria, which carry out the respiratory process [6]. It should be noted that each cell type has its own signature, which is determined not only by the amount of ROS, but also by their composition and the ratio of ROS species to each other. In this regard, the impact of abiotic stressors on different cell types will be different [3]. Long-term exposure to high stress levels leads to the destruction of RNA and DNA and to lipid oxidation [1,7]. Plants have developed a complex enzymatic and non-enzymatic antioxidant system (AOS), which maintains a stable concentration of ROS and prevents excessive ROS accumulation [8,9]. These two systems work together to control ROS levels. The presence of a more powerful AOS system may be one of the mechanisms of plant stress resistance to external adverse effects [9].

Glutathione (GSH), a γ-glutamylcysteinylglycine tripeptide, is an important component involved in many cellular processes in plants [10]. GSH is an unusual peptide that forms a peptide bond between the amino group of cysteine and the carboxyl group of the glutamic acid side chain. GSH is an essential molecule, but there is still a lack of complete clarity on its function. The main function of this small molecule is its antioxidant properties. In addition, GSH has been shown to be a participant in signaling processes in plant responses to abiotic stress [11]. Another function of GSH related to protecting plants from stress is its participation in the detoxification process. GSH is a substrate for the synthesis of phytochelatins, which are a polymeric form of glutathione [12], which is capable of binding heavy metals [13,14]. GSH is also involved in the detoxification of xenobiotics together with glutathione-S-transferase (GST) [15], which catalyzes the formation of a covalent bond between the sulfur atom of the cysteine residue of GSH with an electrophilic compound [16].

GSH reduces ROS and is itself converted into the oxidized form GSSG. Under control conditions, i.e., without stress, the GSH/GSSG ratio is 20:1. Under stress conditions, this ratio changes. The ratio of the reduced form of glutathione to its oxidized form GSH/GSSG shows the level of oxidative stress, which is one of the most important parameters of the cell state.

In plants, GSH effectively protects the cell from the stress-induced formation of reactive oxygen species by binding to ROS [17,18]. An increase in the concentration of hydrogen peroxide is accompanied by the conversion of GSH to the oxidized form GSSG and a change in the GSH/GSSG ratio. From these data, it can be concluded that GSH is a good marker of oxidative stress caused by increased production of H_2_O_2_.

Through immunoprecipitation, it was found that GSH is localized in the nucleus and cyto-sol during the G1 phase of the cell cycle [17,19]. This localization of GSH is assumed to be dynamic. Translocation of GSH from the cytosol to the nucleus during the G1 phase is accompanied by cytosolic oxidation and accumulation of ROS in the cytosol [17,20].

Peptides are localized in all plant organs and cell types [21]. Depending on the localization, peptides exhibit different specific functional activities. Some of these peptides act as short local signaling molecules during plant growth and development, while others are active in environmental adaptation, acting over large distances from the root to the shoot [22,23,24]. The number of functionally characterized peptide hormones exceeds the number of classical plant hormones.

Secreted peptide hormones are divided into three groups: the first group includes peptides that include complex post-translational modifications followed by extensive proteolytic processing; the second group includes peptides rich in cysteine residues and forming disulfide bonds followed by proteolytic processing; and the third group includes peptides also rich in cysteine residues but without proteolytic processing [25].

The most studied peptide hormones are the CLE CLAVATA3 (CLV3)/EMBRYO ENVIRONMENTAL PEPTIDE (CLE) family of peptides, which is associated with stem cell maintenance; PHYTOSULFOKINE (PSK) and PLANT PEPTIDE CONTAINING SULFATED TYROSINE (PSY), which are associated with cell proliferation; ROOT MERISTEM GROWTH FACTOR (RGF, also known as CLE-like or GOLVEN peptide), which is required for the maintenance of the root stem cell niche and transit-enhancing cell proliferation; CASPIAN STRIP INTEGRITY FACTORS (CIFs), which are essential for casparium formation; and RAPID ALKALINATION FACTOR (RALF), which is involved in calcium-mediated signaling and root growth regulation [21]. These short peptides are actively involved in the root development process, especially in the development of primary and lateral roots, as well as root hairs.

Maturation of group 1 peptides, which include CLE, CIF, PSK, PSY, and RGF, occurs as a result of post-translational modification. The most common and important modification is the sulfation of tyrosine residues. Sulfate groups provide stability to the peptide hormone, which is secreted by the apoplast, and specificity for receptor recognition [26]. The introduction of a sulfate group into the peptide structure changes its affinity for proteins that bind to peptides through a sulfur residue, and thus mature peptides already act as signaling ones. Sulfated tyrosine residues, as well as cysteine amino acid residues, can be involved in signaling processes through oxidation–reduction reactions mediated by H_2_O_2_. H_2_O_2_ is the main neutral stable molecule. It can penetrate plant membranes and is considered the predominant ROS involved in intercellular communication [27]. Intercellular communication occurs through a series of reactions in which H_2_O_2_ oxidizes thiol groups (–SH) of cysteine (Cys) residues in proteins. The resulting –SOH oxidative modifications of peptides can transmit the ROS signal further by interacting with other proteins or receptors [28].

The spatiotemporal balance between the maintenance, proliferation, and differentiation of stem cells determines the rate of root growth and is regulated not only by phytohormones, but also by small peptides and ROS [29]. Stem cells in the root meristem are the source of cells involved in the generation of various plant root tissues [30]. Root growth is achieved by a balance of two processes: the maintenance of stem cells surrounding a group of mitotically inactive cells called the quiescent center (QC) and the formation of differentiated cell types. It is known that in the root quiescent center a high oxidative status is maintained [31]. The redox potential of glutathione in these cell types is relatively high, and the GSH:GSSG ratio in the vacuole or endoplasmic reticulum is low [32]. However, in addition to partial cell death at high redox levels, in dormant cells the high redox level affects the processes that determine cell fate [33] and the associated responses to abiotic stress [34].

Tyrosine-sulfated peptides PSK, PSY, and RGF have a positive effect on root development meristems. In addition, PSK promotes the regulation of QC cell division and distal stem cell differentiation [35]. Application of the RGF1 peptide restores the root meristem size defects in the cell division zone of the *tpst-1* mutant. In addition, the root growth defects of *tpst-1* can be completely eliminated in the presence of RGF1, PSK, and PSY1, suggesting that these three peptides are essential for root meristem development [36].

The mechanisms of regulation of root meristem development by the RGF1-receptor signaling pathway have been identified. It has been shown that RGF1, through binding to RGFR, regulates the distribution of ROS in the root development zones of Arabidopsis and increases the stability of the PLT2 protein [37]. It has been suggested that the RGF1 peptide can transmit a signal through intermediate reactive oxygen species, controlling the size of the meristematic zone. After RGF1 treatment, the O_2_^•−^ level in the meristematic zone increases. On the other hand, the concentration of H_2_O_2_ decreases in the elongation and differentiation zone after RGF1 treatment. This distribution of O_2_^•−^ and H_2_O_2_ in the meristematic zone affects its size. It can be concluded that there is a link between ROS and RGF1 signals in the regulation of PLT2 protein stability and root meristem size [37].

One of the largest and best-studied families of peptide hormones is CLE (CLV3/ESR), consisting of 12–13 amino acids [38]. CLE peptides have a wide range of functional activities, including controlling the activity of apical meristems of shoots, roots, and cambium; differentiation of vascular tissues; formation of lateral roots and nodules; early embryogenesis; stomatal development; and response to several environmental factors, such as water availability and changes in the composition of soil nitrogen. Proline residues in CLE peptides are post-translationally hydroxylated and arabinized. Unlike sulfated peptides, CLE peptides negatively affect root meristem development by forming a negative feedback loop between CLE40 and *WUS* (*WUSCHEL*), regulating stem cell proliferation and differentiation processes. It is suggested that two negative feedback loops are formed that control stem cells in Arabidopsis. One loop includes the CLV3 peptide, the other—CLE40 [39].

Previously, it was shown that the short tetrapeptide AEDL stimulates the development of the root system in *Nicotiana tabacumx* [40]. It was suggested that the peptide AEDL acts similarly to the peptide CLV3. The FITC-labeled peptide is localized predominantly in the elongation zone and slightly in the meristem zone. This localization of the peptide AEDL suggests its binding to the hydrophobic leucine-rich motif of the receptor CLV1, thereby preventing its penetration into the stem cell niche [41,42]. Binding of the peptide AEDL to the receptor CLV1 leads to activation of the receptor complex, thereby limiting the stem cell population and activating the stem cell fate determination process. At the same time, the formation of a complex between the peptide AEDL and the receptor CLV1 prevents the penetration of the peptide CLV3 into the meristem from the quiescent zone (QC). By preventing the penetration of the peptide CVL3 into the QC zone, the process of activation of the *Wuschel-like homeobox* (*WUS*) transcription factors and activation of the process of stem cell differentiation occurs.

Since the AEDL peptide is involved in the regulation of the proliferation–differentiation process, and this process depends on the H_2_O_2_/O_2_^•−^ ratio, the aim of this study was to determine the relationship between the AEDL peptide and one of the most important participants regulating the redox balance—GSH.

## 2. Results

### 2.1. Plant Materials

Salt stress is the most common abiotic stress that negatively affects the growth and development of most crop plants [43]. High salt concentrations have a negative effect mainly due to disruption of the ionic and osmotic balance in the cell. In saline soils, high levels of sodium ions lead to delayed plant growth and even death. There are many approaches to reduce the negative effects of salt stress. We have previously shown that growing *N. tabacum* in the presence of the peptide AEDL prevents the negative effects of 150 mM NaCl [40]. It should be noted that the peptide AEDL contributes to a noticeable activation of plant development, especially the root system, compared to the control tobacco (Figure 1, Table 1).

In tobacco grown in the presence of 10^−7^ M AEDL peptide, a slight increase in root length and a more significant increase in shoot height are observed. However, the total crude and dry weight of tobacco exceeds that of the control variant (1.7 times and 1.4 times, respectively). The increase in the total weight of tobacco grown in the presence of AEDL peptide occurs not only due to the shoot, but also due to a powerful root system. The presence of 150 mM NaCl in tobacco (without peptide) leads to a decrease in all morphometric parameters. Morphometric parameters in tobacco grown in the presence of AEDL peptide and 150 mM NaCl also decrease, but they exceed the same values in the control variant (with NaCl).

### 2.2. Expression of RGF1 Gene

Plant growth regulatory factors (RGFs) are specific transcription factors and participate in the regulation of plant root system development [44,45,46]. RGFs are a whole family of the peptides, but the most studied is RGF1.

Peptide AEDL stimulates gene activity in *RGF1* roots by 2.68 times compared to the control *N. tabacum *(Figure 2). Addition of 150 mM NaCl leads to a 1.9-fold decrease in *RGF1* gene activity. Although the presence of AEDL increases *RGF1* gene expression by 1.75 times, its activity does not reach the gene activity in the control *N. tabacum* sample.

### 2.3. ROS Detection

ROS perform both signaling and regulatory functions in plant cells [47]. They are formed in various cell compartments: in chloroplasts, mitochondria, peroxisomes, cytosol, and membranes [48]. Abiotic stresses, including salt stress, lead to oxidative stress and, consequently, to an increase in ROS content [49]. ROS content was determined using the Carboxy-H2DFF marker in vital root cells. Fluorescence intensity varied in different root zones. The distribution of fluorescence in different root zones was assessed by combining images obtained using phase-contrast microscopy and a fluorescent label in one focal plane. Since not all root zones were equally stained for ROS, we assessed the distribution of cells with elevated ROS content in different root zones by combining images taken using phase-contrast microscopy and a fluorescent label in one focal plane. In the control samples of *N. tabacum*, as well as in the samples grown in the presence of AEDL, only a slight staining of ROS production was observed (Figure 3A,B). In tobacco grown in the presence of 150 mM NaCl, an increase in fluorescence intensity was observed, the highest ROS accumulation was found in the zones of elongation and differentiation. In the root cap and in the division meristem zone, the dye was practically not identified. In the differentiation zone, the cells of the peripheral root tissues—the epidermis and cortex—were stained, while the tissues of the central cylinder were practically not stained (Figure 3B,D).

When growing *N. tabacum* in the presence of the peptide AEDL, ROS detection was not identified in the root cap cells and meristem zone, and a slight fluorescent glow was observed in the elongation and differentiation zones. Fluorescence in these zones was most clearly determined in the epidermal cells and to a lesser extent in the cortex cells. When tobacco grown in the presence of the peptide AEDL was exposed to 150 mM NaCl, the ROS content increased compared to tobacco grown without NaCl. The ROS marker fluorescence intensity accumulated in the differentiation and elongation zones (Figure 3). More intense fluorescence of the peripheral root tissues was noted in the cortex and epidermis, and minimal fluorescence was observed in the central cylinder. Thus, the effect of sodium chloride on control tobacco plants differs from the effect of sodium chloride in combination with the peptide AEDL. A distinctive feature of sodium chloride treatment in the presence of the peptide AEDL is a decrease in the proportion of cells stained with the ROS marker in the differentiation zone of tobacco roots.

Thus, if the direct effect of NaCl on tobacco root cells leads to an increase in the number of cells with an increased ROS content in the differentiation and elongation zones (by 10 and 18.2 times, respectively), then in the presence of AEDL, the number of cells with an increased ROS pool in these zones increases significantly less (by 5.8 times). Consequently, the peptide AEDL increases the resistance of cells in the differentiation and absorption zones to stress conditions under salinity (Figure 3). The presence of the peptide AEDL during *Nicotiana tabacum* cultivation protects root tissues, especially epidermal cells.

#### H_2_O_2_ Content

The peroxide content is one of the markers of damage to plant tissues when exposed to stress factors (Table 2).

Based on the obtained data, it can be concluded that 150 mM NaCl leads to an increase in the H_2_O_2_ content in the roots of *N. tabacum* by 1.47 times. Although the H_2_O_2_ content in the leaves is significantly lower than in the roots by 4.45 times, an increase in the concentration of sodium chloride leads to an increase in the H_2_O_2_ content in the leaves by 2.48 times. The peptide AEDL reduces the amount of H_2_O_2_ in the roots of *N. tabacum* by 1.35 times and slightly affects the H_2_O_2_ content in the leaves, reducing its content by 1.08 times. In addition, the peptide AEDL partially neutralizes the negative effect of NaCl, reducing the formation of H_2_O_2_ in the roots by 1.1 times and more significantly in the leaves by 1.52 times.

### 2.4. Antioxidant Activity

Inhibition of the 2,2-diphenyl-1-picrylhydrazyl (DPPH) oxidation process is an indicator of the activity of the antioxidant system. Table 3 presents data on the antioxidant activity (AOA) of *N. tabacum* grown under different conditions and when exposed to 150 mM NaCl.

The AOA in the roots of *N. tabacum* is 1.64 times higher than in the leaves. Salt stress leads to a decrease in AOA in the roots of *N. tabacum* by 3.18 times, and in the leaves only by 1.57 times. Growing *N. tabacum* in the presence of the peptide AEDL increases AOA in the roots by 3.18 times, and in the leaves by 1.57 times. The presence of AEDL leads to an increase in tobacco resistance to the action of 150 mM NaCl and an increase in antioxidant activity. At the same time, AOA in the roots of *N. tabacum* is even slightly higher than in the roots of the control tobacco. The peptide AEDL contributes to an increase in antioxidant activity in the roots of tobacco after exposure to NaCl by 3.32 times and in the leaves by 1.49 times.

#### 2.4.1. Expression of *MnSOD* and *Cu/ZnSOD* Genes

The main participants in the antioxidant system are enzymes—namely superoxide dismutase (SOD) [50]. SODs represent a large family of enzymes that differ in their metal cofactors. The main function of SOD enzymes is to convert superoxide ion (O_2_^•−^) into hydrogen peroxide. Depending on the cofactor, these enzymes differ in their localization.

The expression of *MnSOD* and *Cu/ZnSOD* genes in tobacco roots in the presence of the peptide AEDL increases by 1.85 times and 1.27 times, respectively (Figure 4). An exception is the expression of the *MnSOD* gene in tobacco leaves: the peptide AEDL does not change its level. Addition of 150 mM NaCl leads to a 1.9-fold decrease in the expression level of the *Cu/ZnSOD* gene in *N. tabacum* roots and practically does not change the activity of the *MnSOD* gene. In tobacco leaves, sodium chloride even leads to an increase in the expression of the *Cu/ZnSOD* gene.

#### 2.4.2. SOD Activity in Root of *Nicotiana tabacum*

The activity of superoxide dismutase (SOD) was determined in different zones of the Nicotiana tabacum root based on the ability of superoxide dismutase to inhibit the autooxidation reaction of (R)-4-[1-Hydroxy-2-(methylamino)ethyl]-benzene-1,2-diol in an alkaline medium. The reaction rate is estimated spectrophotometrically by the optical density of the accumulating autooxidation product

In the roots of *Nicotiana tabacum* grown in the presence of the AEDL peptide, an increase in the SOD content is observed, especially in the elongation zone (2.32 μM/mg protein) (Figure 5). The lowest SOD content is in the root cap zone (0.41 μM/mg protein), but it is 1.4 times higher than in the control sample (0.29 μM/mg protein). The presence of NaCl in the nutrient medium leads to a decrease in the SOD content, regardless of the presence of the AEDL peptide. Nevertheless, the SOD content in all zones of the *N. tabacum* root grown in the presence of the AEDL peptide exceeded the SOD values in the control samples.

#### 2.4.3. GSH Content

The GSH content was determined in the roots and leaves of *Nicotiana tabacum* grown on MS medium without and in the presence of the short peptide AEDL (Table 4).

In *N. tabacum* roots, the GSH content exceeds its content in leaves by 2.86 times. However, the GSH content in *N. tabacum* roots can be increased by 3.24 times when growing the plant in the presence of the short peptide AEDL. An increase in GSH concentration was also observed in *N. tabacum* leaves, although to a lesser extent, only by 1.36 times. A high concentration of nitric chloride in the nutrient medium leads to a decrease in the GSH content in tobacco roots by 1.18 times. It is interesting to note that the decrease in GSH concentration in tobacco leaves under the influence of NaCl is more significant, decreasing by 2.15 times. An inverse relationship is observed in tobacco grown in the presence of the peptide AEDL under the influence of NaCl: in leaves, the GSH content decreases by 1.27 times and in roots—by 1.83. It should be noted that although the GSH content in tobacco roots grown in the presence of AEDL decreases under the influence of NaCl, it is not so significant and even exceeds the GSH level in the roots of the control tobacco sample by 1.72 times.

#### 2.4.4. Biosynthesis of GSH, Expression of *GSH1* and *GSH2* Genes

It is known that the biosynthesis of glutathione occurs in two stages. First is the synthesis of γ-glutamylcysteine (γGC) from glutamate and cysteine, which is catalyzed by the enzyme γ-glutamylcysteine synthetase or γ-glutamylcysteine ligase (GSH1, γ-GCL). In the second stage, glutathione synthetase (GS or GSH2) catalyzes the formation of GSH from γGC and glycine [51].
γGCL or GSH1 GS or GSH2

γGlu+ Cys **→** γGluCys+Gly **→** γ GluCysGly (GSH)

When growing *N. tabacum* in the presence of AEDL, the expression of the *GSH1* and *GSH2* genes increases by 1.4 and 1.47 times, respectively (Figure 6). In leaves, the expression of the *GSH1* gene in tobacco in the presence of AEDL increases by 1.29 times, and only by 1.08 times in the presence of the *GSH2* gene. The presence of 150 mM NaCl in the nutrient medium leads to a decrease in the activity of the *GSH1* and *GSH2* genes, especially in the leaves. While the decrease in the activity of the *GSH1* gene in the roots was only 1.14 times, in the leaves, it was 1.83 times. The decrease in the activity of the *GSH2* gene under the influence of NaCl is not as dramatic as that of the *GSH1* gene, the expression level falling by only 1.29 times in the leaves. Growing *N. tabacum* in the presence of AEDL leads to an increase in the expression level of the *GSH1* gene in the roots by 2.68 times, which is even higher than without NaCl by 1.67 times. In tobacco leaves grown under the same conditions, the peptide AEDL also increases the expression of the *GSH1* gene. Although the expression level increases by 2.28 times compared to tobacco leaves grown in the presence of NaCl without the peptide AEDL, the expression level of the *GSH1* gene remains almost at the same level as in tobacco leaves grown in the presence of the peptide AEDL without NaCl.

The addition of NaCl to the nutrient medium slightly reduces the expression of the *GSH2* gene in tobacco roots by 1.05 times and more significantly in leaves by 1.29 times. However, the presence of the peptide AEDL leads to an increase in the activity of the *GSH2* gene in the roots by 1.83 times and especially in the leaves, increasing 2.48 times. It should be noted that the peptide AEDL stimulates the activity of the *GSH2* gene both in the roots and in the leaves of tobacco even more intensely in the presence of NaCl than without it.

The addition of 150 mM NaCl had a negative effect on GSH content (Table 4). The expression activity of the *GSH1* and *GSH2* genes was significantly reduced in both the roots and leaves of *N. tabacum* and its mutant. It is interesting to note that the highest expression of the *GSH1* and *GSH2* genes was in *N.abacumt* grown in the presence of AEDL (even higher than in the mutant), which increased by 2.68 times and 1.83 times, respectively. In *N. tabacum* roots, a slight decrease in GSH content by 1.18 times was observed. But in tobacco grown in the presence of AEDL, the GSH content, although decreased with the addition of 150 mM NaCl, was still increased by 2.03 times compared to the control *N. tabacum* under salt stress. The GSH content in the roots of the tobacco mutant decreases more significantly compared to *N. tabacum* by 2.95 times, and even in the presence of AEDL the concentration was reduced by 1.41 times.

#### 2.4.5. Expression of *GR* and *GST* Genes

GSH binds to ROS, turning into the oxidized form of GSSG, which is reduced by the enzyme glutathione reductase (GR) [52]:GR
GSH + H_2_O_2_ **→** GSSG **→** GSH

Thus, the ratio of the reduced and oxidized forms of GSH:GSSG and, accordingly, the reduced form of GSH, depends on the activity of the GR enzyme.

The *GR* gene activity in tobacco roots is almost independent of the action of sodium chloride, but depends on the presence of the peptide AEDL and increases by 1.41 times. GST is an enzyme that, together with GSH, participates in detoxification (Figure 7). This is especially important under the influence of both abiotic and biotic stresses. The gene activity of *GST* increases in the roots of tobacco grown in the presence of the peptide AEDL by 1.36 times. It should be noted that in tobacco leaves grown in the presence of the peptide AEDL, the expressions of the *GR* and *GST* genes remain virtually unchanged. However, salt stress leads to a sharp increase in the expression of these genes by 4.26 and 4.39 times, respectively.

## 3. Discussion

Plants are constantly exposed to various environmental influences. These influences can be both long-term and short-term and vary in intensity. Under the influence of stress factors, plants either adapt to them or die [43].

Plants have developed a whole complex of counteractions to stressors. Depending on the protective mechanisms, plants under the influence of stress factors either acclimatize or die. Salt stress has a negative effect on the development of *Nicotiana tabacum*: its growth slows down, especially in the root system. The short peptide AEDL promotes the development of the root system of *Nicotiana tabacum* and reduces the negative impact of sodium chloride.

Abiotic stresses, including salt stress, lead to the accumulation of excess ROS. Tobacco grown in the presence of the peptide AEDL accumulates ROS to a lesser extent under the influence of NaCl. Moreover, the distribution of ROS in the zones of the roots of tobacco is different when grown under different conditions. A distinctive feature of *Nicotiana tabacum* grown in the presence of AEDL is that the largest proportion of excess ROS accumulates in the zones of elongation and differentiation and is practically absent in the meristem and cap zones.

The peptide hormone RGF1 belongs to the Root Meristem Growth Factors family and is a secreted peptide of 13 amino acids [44,45,46]. The main function of RGF peptides is to regulate the development of the root system in plants. RGF family peptides are predominantly expressed in the root meristem in the stem cell region and participate in the formation of the root stem cell niche [53]. The RGF1 signaling cascade through receptors regulates the formation of the PLETHORA (PLT) gradient, which is known as the master regulator of root formation [53]. It has been shown that RGF1 is involved in the distribution of ROS along root development zones [54]. RGF1 can transmit a signal through ROS, controlling the size of the meristematic zone. It was found that after RGF1 treatment, the O_2_^•–^ level in the meristematic zone increases, while the H_2_O_2_ content in the elongation and differentiation zones decreases [54].

Based on the obtained data, it follows that the peptide AEDL stimulates the synthesis of the peptide RGF1, the amount of which increases more than 2.5 times. It can be assumed that at the same time there is an increase in the amount of superoxide ion, which can lead to the oxidation of PLT. However, it was found that in the roots of *N. tabacum* in the presence of the peptide AEDL, the expression of the *MnSOD* and *Cu/ZnSOD* genes increases, which promote the conversion of superoxide ion into H_2_O_2_. The increase in the expression of *MnSOD* and *Cu/ZnSOD* genes in the presence of the AEDL peptide is accompanied by an increase in the content of total SOD activity in the roots of *Nicotiana tabacum*. The highest content was noted in the meristem and elongation zones. We have shown that in tobacco roots, the content of H_2_O_2_ in the presence of the peptide AEDL decreases almost 1.5 times. This fact may indicate the active participation of GSH in the neutralization of excess H_2_O_2_, especially since an increase of more than 3 times in the concentration of GSH in tobacco roots in the presence of the peptide AEDL was found.

Glutathione-γ-glutamylcysteinylglycine (GSH), a small molecule, has proven to be an important molecule without which plants cannot develop normally [10]. The reasons why this small molecule is essential are not fully understood, but it can be concluded that GSH performs functions in plant development that cannot be performed by other thiols or antioxidants. The known functions of GSH include roles in biosynthetic pathways, detoxification, antioxidant biochemistry, and redox homeostasis. Since ROS, especially H_2_O_2_, accumulate in plants under various abiotic stresses, many researchers are interested in increasing GSH levels in plants. The main strategies for increasing GSH content are the use of transgenic plants. However, it has been observed that elevated GSH levels do not always have a beneficial effect on plant development.

Motivated by the important role of GSH in plant function, many efforts have been made to increase the GSH content in several plant species. The main strategic approaches to increase GSH content have largely relied on ectopic expression of GCL. Using transgenic plants, GSH content in plants could be increased two to six times [55]. However, many experiments did not achieve the expected results [56]. Overexpression of chloroplast *γ-ECS* in tobacco was accompanied by an increase in GSH levels; however, there was also an increase in oxidation and tissue damage [57]. Another group of researchers found that one of the chloroplast lines with multiple insertions exhibited symptoms of early leaf senescence [58]. Another study showed that the overexpressor experienced a decrease in biomass and photosynthesis [59]. Other authors reported that some transgenic tobacco lines with increased expression and high content of GSH did not show significant impairment of its involvement in primary or defense metabolism, and they consider these tobacco lines to be interesting objects for further studies [18]. Transgenic tobacco plants expressing a more complex StGCL-GS construct were reported to exhibit extreme GSH accumulation (up to 12 μM) in leaves, more than 20–30 times the GSH content of wild-type plants [18]. Surprisingly, this dramatically increased GSH production does not affect plant growth while increasing plant tolerance to abiotic stress. In addition, plants expressing StGCL-GS provide a new, economical source for GSH production that is competitive with existing yeast-based systems [60].

To increase the GSH content, we used the peptide AEDL. When growing tobacco in the presence of the peptide AEDL, the GSH content reliably increased in the roots by 3.24 times and in the leaves by 1.36 times. It should be noted that with this option for increasing the GSH content, the plants feel comfortable, their root systems are more developed compared to the control option, and the leaves have a larger area.

In tobacco grown in the presence of AEDL, the activity of γ-glutamylcysteine ligase increases only 1.47-fold, while *GSH2* expression increases 1.4-fold. The second stage is probably limiting for GSH synthesis. This fact is important for the regulation of GSH content in plants; its accumulation can have negative consequences for normal plant development.

GSH is involved in H_2_O_2_ detoxification in tandem with glutathione peroxidase (GP) and GST. Interestingly, *GP* expression levels were so low that they were not discussed in this study. Probably, the GSH-GST complex played a major role in H_2_O_2_ detoxification and detoxification. This fact is confirmed by the increase in *GR* expression, which is designed to reduce the oxidized form of GSSG to GSH. When growing *N. tabacum* in the presence of the peptide AEDL, a decrease in ROS formation is observed compared to control samples. Low concentrations of the peptide AEDL increase the expression of both *Cu/ZnSOD* and *MnSOD*, as well as the genes responsible for GSH biosynthesis—GSH1 and GSH2—which leads to an increase in AOA.

For a long time, it was believed that the accumulation of ROS had a negative effect, leading to disruption of plant development, tissue damage, and, depending on the degree of the negative impacts, even death. ROS trigger signaling in response to stress, and excess ROS are neutralized by antioxidants to prevent oxidative damage to cells. Recently, accumulating evidence suggests that the redox balance determines the fate of stem cells [61]. Plant growth and development depend on the maintenance and continued differentiation of stem cells located in the quiescent zone (QC) of the apical meristem in roots (RAM) and shoots (SAM). The processes of proliferation and differentiation of stem cells are strictly controlled by signaling molecules, peptides, and transcription factors [42]. Stem cell fate is determined by a negative feedback mechanism between the homeodomain transcription factor *WUSCHEL* (*WUS*), which is expressed in a small subset of organizing center (OC) cells, and the secreted peptide CLAVATA3 (CLV3), which negatively regulates *WUS* expression. Downregulation or loss of WUS function causes plant stem cell shrinkage or death. On the other hand, the peptide CLV3 binds to the CLV1 receptor, a leucine-rich kinase receptor that is located at the boundary of the stem cell niche and interferes with the transit of CLV3 and the suppression of *WUS* activity.

Recently, it was shown that to regulate the processes of proliferation and differentiation of stem cells, a redox balance is necessary, and its main participants are H_2_O_2_ and O_2_^•−^ [61]. Regulation of the redox balance also occurs through a negative feedback loop mechanism. It was found that SOD is localized in the peripheral zone (PZ) and is involved in the conversion of O_2_^•−^ to H_2_O_2_. The accumulation of H_2_O_2_ leads to the suppression of *WUS* expression and an increase in the process of stem cell differentiation. Increasing the content of O_2_^•−^ in the QC leads to an increase in the stem cell niche by activating *WUS* expression. Thus, self-maintenance of the ROS balance in stem cells occurs. There is no evidence of the participation of GSH in the regulation of the redox balance in stem cells; however, we believe that GSH is also integrated into this process.

Increased SODs activity in tobacco in the meristem zone grown in the presence of AEDL peptide results in active neutralization of O_2_^•−^, converting it into H_2_O_2_. A decrease in O_2_^•−^ content in stem cells is accompanied by suppression of *WUS* activity and a decrease in the stem cell pool due to determination of the fate of stem cells in plants [61]. On the other hand, an increase in GSH content and activation of *GR* and *GST* genes involved in detoxification of H_2_O_2_ together with GSH leads to a decrease in H_2_O_2_ content and, accordingly, increased stem cell proliferation and formation of a stem cell pool. Although *N. tabacum* grown in the presence of AEDL peptide also shows a decrease in ROS content and activation of *Cu/ZnSOD* and *MnSOD* genes, the root system becomes more developed compared to the control samples. These results indicate that the peptide is able to participate in the regulation of the redox balance in stem cells. We proposed the following mechanism of stem cell niche regulation (Figure 8).

Figure 8 consists of two loops: the positive RGF1-RGFR1-PLT loop and the negative AEDL-CLV1-*WUS* loop; it also includes a loop of negative regulation of *WUS* activity by H_2_O_2_. RGF peptides are a key factor regulating proximal meristem activity through the PLT pathway [36]. RGF peptides are matured by sulfation of a tyrosine residue by tyrosyl protein sulfotransferase (TPST), whose activity is regulated by auxin [36]. The sulfated RGF1 peptide binds to the RGFR1 receptor. Root meristem maintenance involves MITOGEN-ACTIVATED PROTEIN KINASE (MKK4/5) and MAP KINASE (MPK3/6) as downstream signaling components of RGF1-RGFR1, modulating the expression of transcription factors PLT1 and PLT2 [62,63]. Thus, the peptide forms a positive RGF1-RGFR1-PLT loop, activating the formation of stem cell niches. It is known that GSH prevents the oxidation of PLETHORA, thereby demonstrating the active participation of GSH in this process [64,65,66]. GSH helps to prevent the oxidation of the transcription factor PLT and promotes the neutralization of H_2_O_2_. Increased H_2_O_2_ suppresses the expression of *WUS*, activating the process of stem cell fate determination [36,64,65]. It is known that the 13-membered peptide CLE40 negatively regulates the expression of *WUS*, forming a CLE40-CLV1-*WUS* loop. By binding to the CLV1 receptor, the CLE40 peptide does not penetrate the QC zone and thus contributes to the inhibition of *WUS* activity. Inhibition of *WUS* activity contributes to the activation of the process of determining the fate of stem cells in the apical zone of the root [41,42]. The AEDL peptide is able to bind to the CLV1receptor, which is rich in leucine repeats. It was previously shown that FITC-AEDL does not penetrate the meristem zone and, at the same time suppresses, *WUS* activity [40]. Based on this, we previously proposed a scheme in which the AEDL peptide, like the CLE40 peptide, forms a negative feedback loop and participates in the regulation of the process of determining the fate of stem cells [40].

AEDL stimulates the activity of GSH synthesis. A high redox balance is necessary for normal stem cell homeostasis [61]. High glutathione concentrations can disrupt this balance and, as a result, lead to significant changes in plant development. It can be suggested that the GSH tripeptide may bind to the CLV1 receptor, which helps prevent its entry into the QC and neutralize H_2_O_2_. An increase in H_2_O_2_ suppresses *WUS* expression, activating the process of stem cell fate determination. It can be speculated that AEDL and CLV1 prevent GSH entry into the QC and neutralize H_2_O_2_. Thus, we suggest the participation of GSH in two feedback loops, RGF1-RGFR1-PLT and AEDL-CLV1-*WUS*, thereby actively participating in the formation of the stem cell niche and determination of the stem cells’ fates.

## 4. Materials and Methods

### 4.1. Plant Material

Tobacco seeds (*Nicotiana tabacum* L.) of the Samsun variety were placed in flasks containing hormone-free Murashige–Skoog (MS) medium [40]. In parallel, 150 mM NaCl or 10^−7^ M AEDL or 10^−7^ M AEDL and 150 mM NaCl were added to the medium. After 28 days, tobacco seedlings grown under the different conditions were collected, and morphometric parameters were determined. The roots were collected and used for further measurements. The AlaGluAspLeu peptide was synthesized by IQChem (St. Petersburg, Russia). The experiments were carried out in triplicate.

### 4.2. ROS Determination

To determine ROS by the fluorescent method, root tips (4–5 mm) of seedlings were incubated in 25–50 nM carboxy-H2DFFDA (Thermo Fisher Scientific, Waltham, MA, USA) for 30 min according to the method in [40]. The samples were analyzed under an Olympus BX51 fluorescent microscope (Tokyo, Japan) with a 10× objective at a wavelength of 490 nm. Images were obtained using a Color View digital camera (Munster, Germany). ImageJ software (version 1.54i) was used to measure fluorescence intensity.

### 4.3. Biochemical Analysis

Antioxidant activity (AOA) was determined by the decrease in the coloration of the 5 × 10^−5^ M alcohol solution 2,2-diphenyl-1-picrylhydrazyl (DPPH). Absorbance was measured at λ = 517 nm. AOA was calculated using the following formula: (Ao − A/Ao) × 100% [66]. The concentration of peroxide in aqueous extracts of plant material was determined by the reduction in coloration of a 0.02 M solution of KMnO_4_. Absorbance was measured at λ = 480 nm [67]. The glutathione (GSH} content in mkM/g plant biomass was determined by the Elman method by the appearance of color after the addition of 0.01 M 5,5′-Dithio-Bis-(2-Nitrobenzoic Acid) alcohol solution. Absorbance was measured at λ = 412 nm [68]. To determine the activity of superoxide dismutase (SOD), 0.1 g of *Nicotiana tabacum* roots were homogenized in 0.2 M Na_2_CO_3_ (pH 10.3). The activity of superoxide dismutase (SOD) was determined by measuring the inhibition of the autoxidation reaction of 18.2 μM (R)-4-[1-Hydroxy-2-(methylamino)ethyl]-benzene-1,2-diol in 0.2 M Na_2_CO_3_ after 10 min of incubation at 30 °C [69]. The activity was measured spectrophotometrically by the change in absorbance at 347 nm. The activity was calculated (Ex-Eo/Eo/mg protein/min).

### 4.4. Total RNA Isolation and Gene Expression Analysis

Using a standard RNA isolation kit-Extran RNA Syntol (Moscow, Russia), total RNAs were isolated from *Nicotiana tabacum* roots and shoots grown under different conditions. cDNAs were synthesized by reverse transcription according to the standard method (Syntol, Moscow, Russia).

RT-PCR using SYBR Green I (Syntol) was performed in a CFX 96 Real-Time thermal cycler (BioRad, Berkeley, CA, USA). Information on the structure of the *FeSOD*, *MnSOD*, *GSH1*, *GSH2*, *GR*, and *GST* genes in *N*. *tabacum* was obtained from NCBI. Primers for the genes were synthesized by Syntol. *GaPDh* gene was used as a reference gene. Each RT-PCR reaction was performed in three repeats.

### 4.5. Statistical Methods

Statistical processing of experimental data was carried out using one-way analysis of variance (ANOVA) and Student’s *t*-test (R version 4.3.1) with significant differences at *p* < 0.05. The least significant difference method was used to test significance. Values are presented as means ± standard deviations of triplicate biological replicates

## 5. Conclusions

It is noted that the AEDL peptide stimulates plant growth, especially in the root system. The central dormant zone responsible for plant development has a high oxidative level, which regulates the fate of stem cells. It is assumed that GSH and the AEDL peptide form an additional negative feedback loop, participating in the regulation of the redox balance in the stem cell niche and the regulation of the fate of stem cells. It was found that AEDL activates GSH biosynthesis. From the presented scheme, it follows that high concentrations of GSH can lead to disruption of this balance and, as a consequence, to significant changes in the process of plant development. However, AEDL, controlling the binding of GSH to the CVL1 receptor, prevents the penetration of excess GSH into the meristem zone and thereby prevents a decrease in the redox balance.

## Figures and Tables

**Figure 1 ijms-26-00289-f001:**
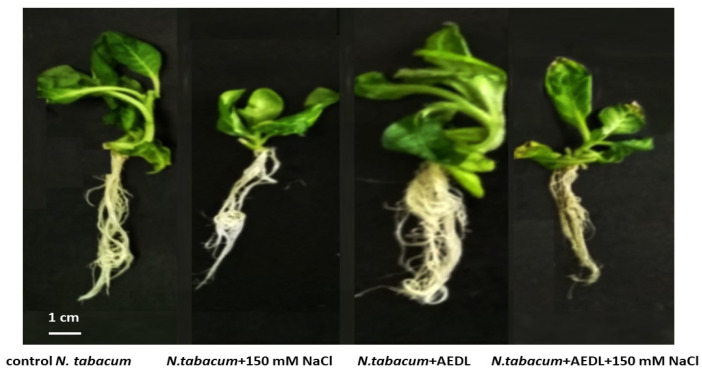
*Nicotiana tabacum*, grown in different conditions. Data were expressed as mean ± standard deviation (SD; n = 30).

**Figure 2 ijms-26-00289-f002:**
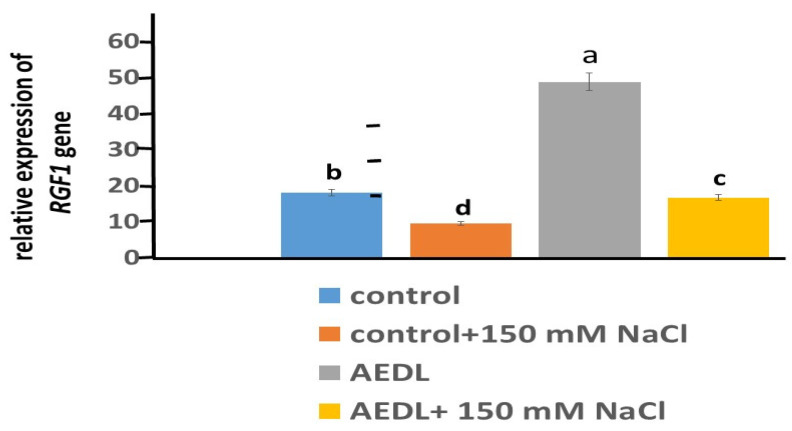
Expression *RGF1* gene in root *Nicotiana tabacum*. Data are expressed as mean ± standard deviation (n = 3); a–d—indicate significant difference using Student’s *t*-test (*p* < 0.05).

**Figure 3 ijms-26-00289-f003:**
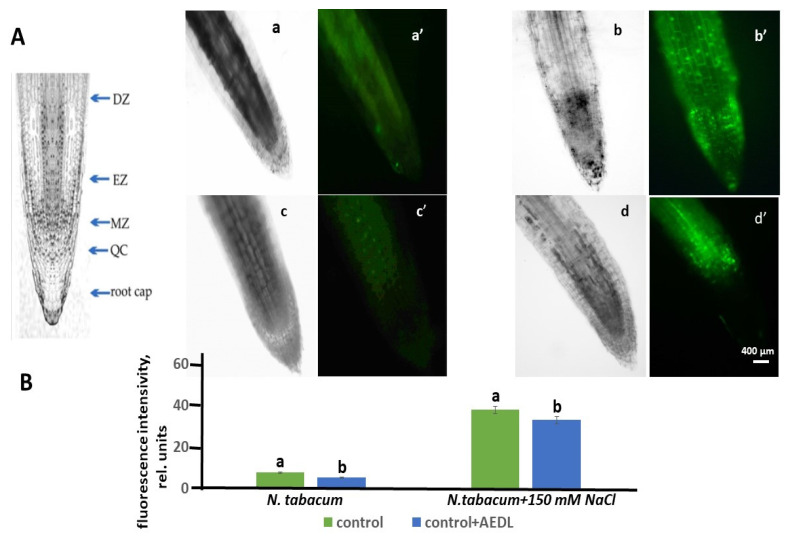
(**A**) Distribution of ROS+ and ROS− cells in the root zones *N. tabacum*. Control—(**a**,**a’**); 150 mM NaCl—(**b**,**b’**); AEDL—(**c**,**c’**); AEDF + 150 mM NaCl—(**d**,**d’**). DZ—differentiation zone; EZ—elongation zone; MZ—meristem zone; QC—quiescent center. Bar 400 µm. (**B**) The intensity of ROS fluorescence in *N. tabacum*, grown in different conditions. (**C**) Distribution of fluorescence intensity in the root zones. (**D**) Distribution of fluorescence intensity in the root tissues. Data are expressed as mean ± standard deviation (n = 3); a–c—indicate significant difference using Student’s *t*-test (*p* < 0.05).

**Figure 4 ijms-26-00289-f004:**
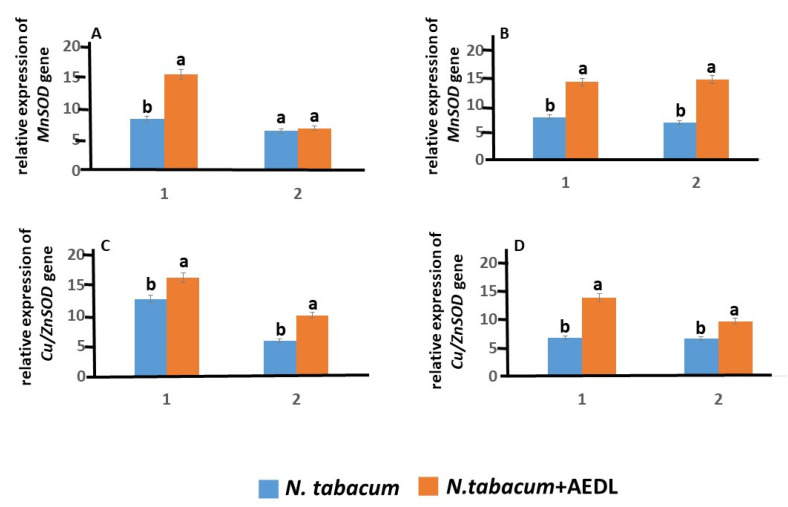
Expression of *MnSOD* and *Cu/ZnSOD* genes in *N. tabacum* in roots (1) and leaves (2), grown in different conditions: (**A**,**C**) control; (**B**,**D**) 150 mM NaCl. Data are expressed as mean ± standard deviation (n = 3); a,b—indicate significant difference using Student’s *t*-test (*p* < 0.05).

**Figure 5 ijms-26-00289-f005:**
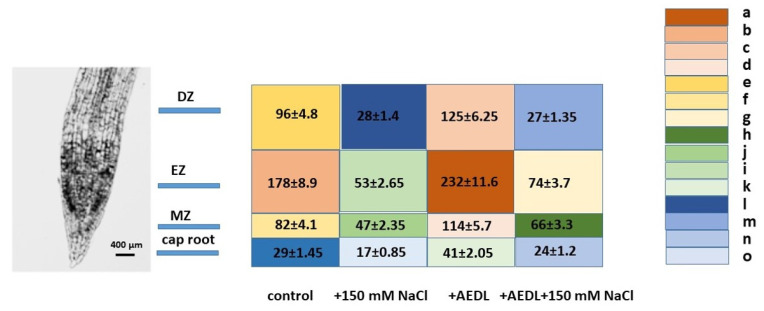
Distribution of SOD activity (nM/mg protein/min) in root zones of *Nicotiana tabacum.* MZ—meristem zone; EZ—elongation zone; DZ—differentiation zone. Data are expressed as mean ± standard deviation (n = 3); a–o—indicate a significant difference using Student’s *t*-test (*p* < 0.05).

**Figure 6 ijms-26-00289-f006:**
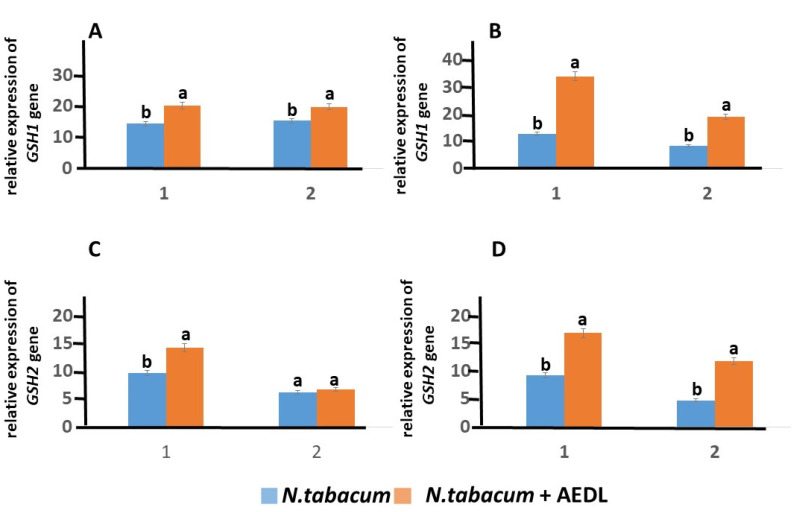
Expression of the *GSH1* and *GSH2* genes in *N. tabacum* in roots (1) and leaves (2), grown in different conditions: (**A**,**C**) control; (**B**,**D**) 150 mM NaCl. Data are expressed as mean ± standard deviation (n = 3); a,b—indicate significant difference using Student’s *t*-test (*p* < 0.05).

**Figure 7 ijms-26-00289-f007:**
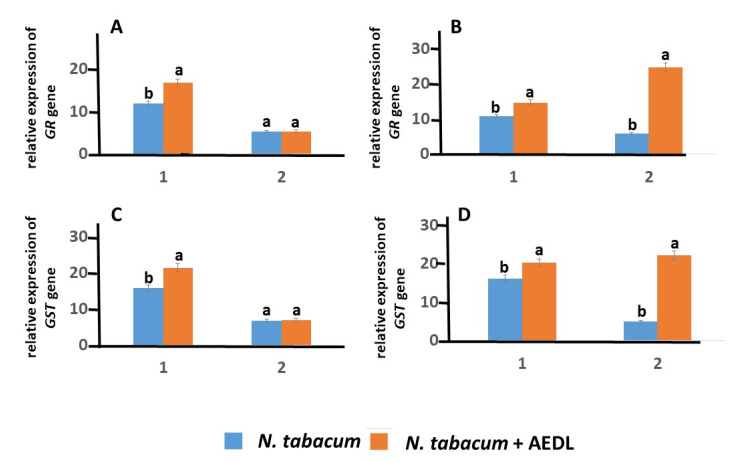
Expression of the *GR* and *GST* genes in *N. tabacum* in roots (1) and leaves (2), grown in different conditions: (**A**,**C**) control; (**B**,**D**) 150 mM NaCl. Data are expressed as mean ± standard deviation (n = 3); a,b—indicate significant difference using Student’s *t*-test (*p* < 0.05).

**Figure 8 ijms-26-00289-f008:**
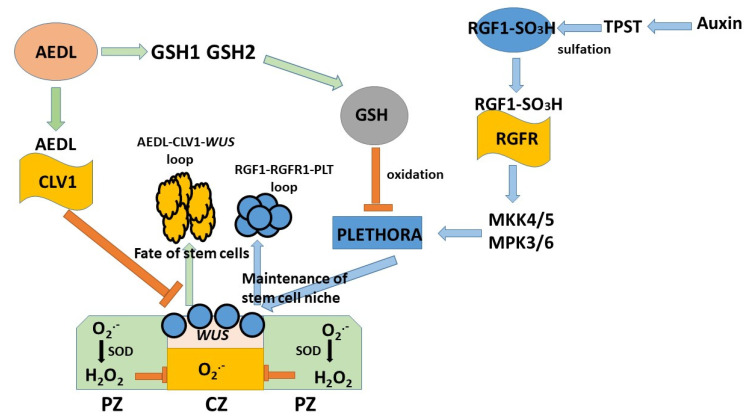
RGF1 peptide and GSH form a positive feedback loop that activates stem cell proliferation. AEDL peptide and *WUS* form a negative feedback loop that activates stem cell differentiation.

**Table 1 ijms-26-00289-t001:** Morphometric parameters of tobacco seedlings growth under different condition.

Variant	Root Length, cm	Shoot Height, cm	Crude Weight, g	Dry Weight, g
control	5.5 ± 0.27 a	3.7 ± 0.18 b	1.5 ± 0.07 b	0.41 ± 0.02 b
+150 mM NaCl	3.8 ± 0.19 c	2.2 ± 0.11 d	0.92 ± 0.04 d	0.37 ± 0.01 c
+AEDL	5.8 ± 0.29 a	4.8 ± 0.24 a	2.5 ± 0.12 a	0.58 ± 0.03 a
+AEDL + 150 mM NaCl	4.8 ± 0.24 b	3.1 ± 0.15 c	1.3 ± 0.06 c	0.44 ± 0.02 b

Weight—total weight of root and shoot. Data were expressed as mean ± standard deviation (SD; n = 3); a–d—indicate significant difference using Student’s *t*-test (*p* < 0.05).

**Table 2 ijms-26-00289-t002:** H_2_O_2_ content in *N. tabacum*, grown in different conditions.

Varieties	Growth Condition	H_2_O_2_ µM/g
*N. tabacum* root	control	12.1 ± 0.60 c
	+AEDL	8.93 ± 0.45 d
	+NaCl	17.8 ± 0.89 a
	+AEDL + NACL	16.1 ± 0.80 b
*N. tabacum* shoot	control	2.72 ± 0.14 c
	+AEDL	2.51 ± 0.12 c
	+NaCl	7.98 ± 0.40 a
	+AEDL + NACL	5.23 ± 0.26 b

Data are expressed as mean ± standard deviation (SD; n = 3); a–d—indicate significant difference using Student’s *t*-test (*p* < 0.05).

**Table 3 ijms-26-00289-t003:** AOA in *Nicotiana tabacum*, grown in different conditions.

Varieties	Growth Condition	AOA %, Inhibitory
*N. tabacum* root	control	37.91 ± 1.89 b
	+AEDL	53.70 ± 2.68 a
	+NaCl	11.90 ± 0.59 c
	+AEDL + NACL	38.10 ± 1.90 b
*N. tabacum* shoot	control	23.13 ± 1.16 b
	+AEDL	28.40 ± 1.42 a
	+NaCl	14.71 ± 0.73 d
	+AEDL + NACL	21.96 ± 1.10 c

Data are expressed as mean ± standard deviation (SD; n = 3); a–d—indicate significant difference using Student’s *t*-test (*p* < 0.05).

**Table 4 ijms-26-00289-t004:** GSH content in *N. tabacum* grown in different conditions.

Varieties	Growth Condition	GSH, µM/g
*N. tabacum* root	control	0.80 ± 0.04 c
	+AEDL	2.59 ± 0.13 a
	+NaCl	0.68 ± 0.03 d
	+AEDL + NACL	1.38 ± 0.07 b
*N. tabacum* shoot	control	0.28 ± 0.01 c
	+AEDL	0.38 ± 0.02 a
	+NaCl	0.13 ± 0.01 d
	+AEDL + NACL	0.30 ± 0.01 b

Data are expressed as mean ± standard deviation (SD; n = 3); a–d—indicate a significant difference using Student’s *t*-test (*p* < 0.05).

## Data Availability

Data are contained within the article.

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
