# Peer review of "Peptide AEDL and Glutathione Stimulates Root Development Nicotiana tabacum"

_ijms, 2024, doi:10.3390/ijms26010289_

Round 1
Reviewer 1 Report (Previous Reviewer 1)
Comments and Suggestions for Authors
The authors have made a valuable attempt to improve the manuscript, but there are still major flaws that invalidate the results.
The petides' phenotype is not convincing because it only provides a picture of a single plant and lacks information on fresh weight, dry weight, growth, or elongation.
Figure 5 speaks about SOD content, and it should speak about SOD activity. Also, the units (mkM/mg protein) are difficult to understand, as the standard is the use of nmols of reactive per mg protein per minute.
GR and GST activity has not been determined, so one cannot include these enzymes in the model as expression by qPCR does not indicate the activity directly.
Another major problem is the description of the gene expression analysis. Which genes serve as references in the RT-PCR experiments? This information is essential.
Author Response
Thank you for your careful review of the manuscript.
The authors have made a valuable attempt to improve the manuscript, but there are still major flaws that invalidate the results.
The petides' phenotype is not convincing because it only provides a picture of a single plant and lacks information on fresh weight, dry weight, growth, or elongation.
Reply: Thank you for the recommendation. We have entered the data of morphometric parameters of tobacco grown in different conditions.
Figure 5 speaks about SOD content, and it should speak about SOD activity. Also, the units (mkM/mg protein) are difficult to understand, as the standard is the use of nmols of reactive per mg protein per minute.
Reply: Thank you for your comment. We have presented the data in a new format in nM/mg protein/min. We have also edited the method for determining SOD activity.
GR and GST activity has not been determined, so one cannot include these enzymes in the model as expression by qPCR does not indicate the activity directly.
Reply: We agree with you. The activity of enzymes is not directly determined by qPCR. Therefore, we did not include these enzymes in the model, but only mentioned them in the discussion.
Another major problem is the description of the gene expression analysis. Which genes serve as references in the RT-PCR experiments? This information is essential.
Reply: In reaction RT-PCR reaction we used GaPDh gene as a reference.
We have entered these data in the methods section.

Reviewer 2 Report (Previous Reviewer 2)
Comments and Suggestions for Authors
Thank you for revison.It is rather cosmetic changes, many more imporvemnet still need. Major questions are to M&M part. There are link tomprevious paper, which did not provide enough details.
Model require more clarity and more evidences.
Lines 14- 15: „However, an increase in the GSH content in plants can negatively affect…“ this statemnet did not confirm in abstarct.
Line 33: „exposure to oxidative stress“ ?? exposure to stressfull conditions.
Line 36: „their accumulation of ROS“ ?? of ROS = redudndant.
Moreover, author miss real mechnansim of stress exposure effect: it is disbalance between different cell type/organs. Each cell type respond to stressfull conditions differently because they have different RNA and epigonome profile. This, in turn, les to disbalane between developmnet and affected plant growth.
Lipid peroxidation is a terminal stage of the stress.
Line 53: „the oxidized form of GSSG“ = „the oxidized form GSSG“.
It seems that authors do not take into account chemical properties of GSH. For oxidation of GSH to GSSG one require a huge amopunt of ROS (H2O2), while at least t0 fold less ROS require for ASC oxidation. So, GSH can not be consider as main antooxidant, despite of some publications.
Line 62: GSH can be consider as a marker only fro very high ROS level (near damage one).
Line 65: “fase G1 cell cycle” ¿?? G1 phase of cell cycle.
Queval, G.; Thominet, D.; Vanacker, H.; Miginiac-Maslow, M.; Gakière, B.; Noctor, G. H2O2-activated up-regula- 680
tion of glutathione in Arabidopsis involves induction of genes encoding enzymes involved in cysteine synthesis 681
in the chloroplast. Molecular Plant, 2009, 2, 344–356
Line 105: „stem cells cells“ ?? Moreover, which stem cell do you mean? Root stem cell form after root primordia induction and serve as end point of the polar auxin transport. While shoot stem cell (SAM) is a auxin starting point. Plesae, clarify.
Line 150: „stem cell differentiation“ – plesae, clarify what do you mean as this term? Histomne de.-acetylation? Cell cycle ? In whcih cell? How much stem cell in tobacco root?
Line 157: „redox balance“ ?? Please, clarify what do you mean. ASC/DHA? HADPH/NADP?? GSH/GSSG? Redox balance itself has no sense without definition which one and location.
line 187: plasma membrane, and cell membrane??? What the differences??
Line 189: “ROS content was determined using the Carboxy-H2DFF marker. ROS production was detected in all root tissues, but with different fluorescence intensities in different root zones. Since not all root zones were equally stained for ROS, we assessed the distribution of cells with elevated ROS levels in different root zones“ ?? Fluorence level can not be marker of ROS balance (not contents) in zone. Different cell typehave different size (arae in your case) and fluorescence dependent form area, vaciole soze etc. Moreover, ROS production can be neasure only if you completely stop scavenging process. But you did not do this. Please, clarify in the text.
Line 200: “the epidermis and cortex - were stained“ ?? which cortex? Tobacco have three cortex layers, exodermis and epidermis.
Line 212: „ROS marker“ ?? maybe you mean rOS detection?
Line 219: “in the cortex” ¿? Which cortex layer? 1, 2 or 3??
Peroxide contents ? Maybe you mean balance?? It is not a marker of damage.
Table 1: How it was measured?? mkM =µM.
Lines 301: „Content of SODs „ ???? Activity?
The methods is unclear. Citation 65 did not provide precise protocol , have a very few citation etc. You need to provide details in M&M part.
There are also a question about “contents”. If plants have a different morphology (more compact cell under NaCl) it is necessary to measure contents per unit(cell), not per FW.
Line 316: please, provide details how did you measure SOD contents and how it link with SOD activity.
Line 471: „high GSH content did not show significant impairment in functional activity“ which functional activity?
Line 514: „proliferation and differentia- 514 tion of stem cells, a redox balance is necessary and its main participants are H2O2 and O2“ which redox balance? ASC? NADPH? What is proliferation and differentiation of stem cell?
Model very speculative. SOD “contents” is unclear, including methods. Clear balance of superoxiude (DHE localization) need to be provided for conclusion. Buffer/watercan not be used forROS determination since they are players itself.
LINE 588: ROS determination, not fluorescence microscopy.
Moreover, it will be great to mention here at least some details. As it is clear from citation 40, ROS measure is not optimal. Authors incubated samples in water 30 minutes and washed with water what include number of stress (ions balance, mechanical stress during mounting). This is will lead to changes in ROS.
Line 599: „concentration of peroxide in aqueous solutions of plant material“ ??? completely unclear. What is aqueos solution? How it was done?
Author Response
Thank you for your careful review of the manuscript.

Round 2
Reviewer 1 Report (Previous Reviewer 1)
Comments and Suggestions for Authors
In this version authors have amended some of the major issues issues raised in the previous round of review. Now the results are more credible with the phenotypic data.
Another problem is using only one gene as reference for RNA seq experiments. This is not accepted in many journal, so please, consider this in future experiments.
Before acceptance, please, in each figure, include which statistical analysis have you performed to calculate the ANOVA.
Author Response
Thank you for your valuable comments.
In this version authors have amended some of the major issues issues raised in the previous round of review. Now the results are more credible with the phenotypic data.
Another problem is using only one gene as reference for RNA seq experiments. This is not accepted in many journal, so please, consider this in future experiments.
Reply: Thanks for the recommendations. In this case, we measured only relative expression, our goal was not to determine the amount of PCR products.
Before acceptance, please, in each figure, include which statistical analysis have you performed to calculate the ANOVA.
Reply: Thanks for the comment. We have added captions under the figures and tables.
Reviewer 2 Report (Previous Reviewer 2)
Comments and Suggestions for Authors
Thank youfir partially answers.
Authirs need to carefully read whole text. correct some citations which did nit fit with contents, explain how they measure SOD from root section with size 200 µm etc-
Line 34: „the ratio of ROS to each other“ ???
Line 36: not long-term, but high stress level.
Citations 19 and 20 did not fit with the contents, as well as some others. Need to be clarify all citations.
Line 114: „GSH:GSSG ratio in the vacuole“ ?? the original text itself rise a lot questions about relaibility. Moreover, there atre mayn types of vacuoles in root: alfa, betta and gamma. Which do you mean?? PSV? Lytic?
Line 115: „exposure to high redox potential“??? you can not exposure to potential.
Line 127: citation 37: despite paper published, there is no any formazan have been detected as superoxide detection. That’s why we can not tell about ROS balance.
Line 159: “depends on the H2O2/O2.- ratio,” - per cell? How did you confirm this in cellular level?
Line 210: “ROS production was detected in all root tissues, but with different fluorescence intensities in different root zones“?? Unclear, require clarification.
„the highest intensity of ROS production“ ??? Production can ntnbe measured, you measure accumulation. Moreover, itos not a ROS, but H2O2 under stress conditions.
Line 322: “The content of superoxide dismutase” ¿? Activity??
Line s491- 492: grammarly,
Line 513: “GSSH to GSH.” ¿????
Line 545: “in active neutralization of O2.-, converting them into H2O2.” ¿? No direct evidences provided. Speculation
Fate of stem cell? Please, rovide claer definion.
Stem cell formation occurred inthe embryos and during earky seeds germination. In this case you can tell about maintenance.
Line 584: 61 is about SAM, not RAM Data can not be compared.
Line 605: „our method [40].“ – redare should not search for older publications. Descriptions must be here.
Line 618 – 622: how did you collect 100 mg form QC? EZ??
Comments on the Quality of English LanguageMany senteces require grammar correcvtions
Author Response
Thank you for your valuable comments.
Thank youfir partially answers.
Authirs need to carefully read whole text. correct some citations which did nit fit with contents, explain how they measure SOD from root section with size 200 µm etc-
Reply: Thank you for your comment. The text has been checked and corrected. The root zones were separated under a microscope with a magnification of x100. The root zones are clearly visible. The quiescent zone was combined with the meristem zone.
Line 34: „the ratio of ROS to each other“ ???
Reply: In the text we specified “ROS species”
Line 36: not long-term, but high stress level.
Reply: Thanks for the recommendation. Added “high stress level”
Citations 19 and 20 did not fit with the contents, as well as some others. Need to be clarify all citations.
Reply: Sorry, we made changes
Line 114: „GSH:GSSG ratio in the vacuole“ ?? the original text itself rise a lot questions about relaibility. Moreover, there atre mayn types of vacuoles in root: alfa, betta and gamma. Which do you mean?? PSV? Lytic?
Reply: Yes, we agree that in roots (and not only) there are several types of vacuoles (but not many).
Although the authors of the publication do not specify in which type of vacuoles the GSH:GSSG ratio is low, we can assume that we are talking about PSV vacuoles.
Line 115: „exposure to high redox potential“??? you can not exposure to potential.
Reply: Thanks for the comment, we have edited it
Line 127: citation 37: despite paper published, there is no any formazan have been detected as superoxide detection. That’s why we can not tell about ROS balance.
Reply: Thanks for the comment,
Line 159: “depends on the H2O2/O2.- ratio,” - per cell? How did you confirm this in cellular level?
Reply: We do not confirm this in this manuscript, this is literary data [61]
Line 210: “ROS production was detected in all root tissues, but with different fluorescence intensities in different root zones“?? Unclear, require clarification.
Reply: Thanks for the comment, we have edited it
„the highest intensity of ROS production“ ??? Production can ntnbe measured, you measure accumulation. Moreover, itos not a ROS, but H2O2 under stress conditions.
Reply: The marker Carboxy-H2DFF stains all ROS species. H2O2 also belongs to ROS.
Line 322: “The content of superoxide dismutase” ¿? Activity??
Reply: Thanks for the comment, we have edited it
Line s491- 492: grammarly,
Reply: Thanks for the comment, we have edited it
Line 513: “GSSH to GSH.” ¿????
Reply: Thank you. Sorry for the typo.
Line 545: “in active neutralization of O2.-, converting them into H2O2.” ¿? No direct evidences provided. Speculation
Reply: Of course, the peptide is not directly involved in the conversion of O2- into H2O2, but indirectly, through stimulation of GSH biosynthesis.
Fate of stem cell? Please, rovide claer definion.
Reply: QC contains 4 types of initials: columella, epidermis, cortex and central cylinder. Plants maintain populations of pluripotent stem cells in RAM (SAM), which continuously produce new aboveground organs.
Stem cell formation occurred inthe embryos and during earky seeds germination. In this case you can tell about maintenance.
Reply: Thanks for the comment. More correctly - maintenance
Line 584: 61 is about SAM, not RAM Data can not be compared.
Reply: The structure of the quiescent center in SAM and RAM are similar, so we compared them. But in the article [61] the authors compared SAM and RAM
Line 605: „our method [40].“ – redare should not search for older publications. Descriptions must be here.
Reply: We have added to the description of the method and added a reference
Line 618 – 622: how did you collect 100 mg form QC? EZ??
Reply: Of course, we did not collect QC directly. We combined this zone with the meristem zone.
Round 3
Reviewer 2 Report (Previous Reviewer 2)
Comments and Suggestions for Authors
Thank you! Some points still corrections.
My best regards.
"accumulation of ROS production" ??? = ROS accumulation.
Line 62: "the GSH/GSSG ratio is almost equal to 1." do you mean GSH = GSSG.
Line 69: citation 20: very interesting determination of GST activity in the nucleus in G1 and usage of the not-described method of H2O2 measurement.
Line 114: "the GSH:GSSG ratio in the vacuole" ? PSV mean protein storage vacuole is only in QC. Maybe you mean lytic vacuole (80-90% cell have lytic vacuole). ??
Line 797: "62. 115. Shao, Y.; Y...- "?? what is 115???
Line 159: Citation 61 was wrong. You need to cite at least Yamada et al., 2020. RGF1 controls root meristem size through ROS signalling | Nature
It is very essential to know that authirs did not quantify ROS level per cell and did not even detect ROS as formazan. They measured phenolic compounds instead of superoxide.
"QC contains 4 types of initials: columella, epidermis, cortex and central cylinder." ???? QC contains cortex/endidermis initials; Epidermis cell and columella origanted from CSC. Pericycle/ vasculature have a separate initials.
Author Response
Thank you for such a thorough review of the manuscript. Your advice was very professional and very valuable in this particular area of ​​research
Thank you! Some points still corrections.
My best regards.
Thank you!
"accumulation of ROS production" ??? = ROS accumulation.
Reply: Corrected
Line 62: "the GSH/GSSG ratio is almost equal to 1." do you mean GSH = GSSG.
Reply: Indeed, we meant that they are almost equal. We have edited this sentence
Line 69: citation 20: very interesting determination of GST activity in the nucleus in G1 and usage of the not-described method of H2O2 measurement.
Reply: Sorry, this quote does not describe the determination of GST activity in the nucleus in G1 and the use of an unreported method for measuring H2O2.
Line 114: "the GSH:GSSG ratio in the vacuole" ? PSV mean protein storage vacuole is only in QC. Maybe you mean lytic vacuole (80-90% cell have lytic vacuole). ??
Reply: Perhaps.
Line 797: "62. 115. Shao, Y.; Y...- "?? what is 115???
Reply: Sorry, Corrected
Line 159: Citation 61 was wrong. You need to cite at least Yamada et al., 2020. RGF1 controls root meristem size through ROS signalling | Nature
Reply: Thanks for the reference. We have added this reference and others to the text.
It is very essential to know that authirs did not quantify ROS level per cell and did not even detect ROS as formazan. They measured phenolic compounds instead of superoxide.
Reply: We measured superoxide using the method patented by Sirota T.V.
"QC contains 4 types of initials: columella, epidermis, cortex and central cylinder." ???? QC contains cortex/endidermis initials; Epidermis cell and columella origanted from CSC. Pericycle/ vasculature have a separate initials.
Reply: Thanks for the detailed description
Round 4
Reviewer 2 Report (Previous Reviewer 2)
Comments and Suggestions for Authors
line 62: at low stress GSH represent 90% andbGSSG less as 10n%mor even lower. Equal they became at high stress.
Line 69: yes, they did not mentions this directly, but dye they use became a fluorescent only in the presence of GST. So, they measure GST localisation, not GSH itself, If GST is absent in cytoplasm, of course, no signal can be there.
GSH presence in vacuaoleis very questionable because essential is distribution among vacuole type. So far there is nt relaible methods of isolation "intact" vacuole" which can not leak GSH.
Onlyin situ methodsis possible, but majority of GSH sentitive dye require GST activity. Moreover, lytic vacole have vaery low pH what alos can affect fluo signal. To avoid confussion it is better to avoid this part, if possible.
Citation 66 is about phenolic componds, not about superoxide. Evene despite they calim superoxide. dx.doi.org/10.17504/protocols.io.bx49pqz6
Please, be carefully with this.
My best regards!
Author Response
Thanks for the recommendations and protocol.
line 62: at low stress GSH represent 90% and GSSG less as 10n%mor even lower. Equal they became at high stress.
Reply: Thank you, edited
Line 69: yes, they did not mentions this directly, but dye they use became a fluorescent only in the presence of GST. So, they measure GST localisation, not GSH itself, If GST is absent in cytoplasm, of course, no signal can be there.
GSH presence in vacuaoleis very questionable because essential is distribution among vacuole type. So far there is nt relaible methods of isolation "intact" vacuole" which can not leak GSH.
Onlyin situ methodsis possible, but majority of GSH sentitive dye require GST activity. Moreover, lytic vacole have vaery low pH what alos can affect fluo signal. To avoid confussion it is better to avoid this part, if possible.
Reply: GSH is detected in all compartments by immunolabeling. Queval found GSH in vacuoles. GSH binds to electrophilic compounds and transports them to the vacuole for further processing. But we agree with you that the concentration of GSH is too low to reliably detect it in vacuoles. We removed this from the text and left only the localization in the cytosol
Citation 66 is about phenolic componds, not about superoxide. Evene despite they calim superoxide. dx.doi.org/10.17504/protocols.io.bx49pqz6
Reply: We agree with you. However, we have provided this reference regarding the scheme of participation of the peptide and ROS in the positive activation of QC maintenance.
Thank you very much for the protocol. We will definitely use it.
Please, be carefully with this.
My best regards!
Round 5
Reviewer 2 Report (Previous Reviewer 2)
Comments and Suggestions for Authors
THANK you. The text is OK. Minor polishing are required.
Author Response
Thank you, your recommendations were professional and friendly, and helped us edit the manuscript correctly.
This manuscript is a resubmission of an earlier submission. The following is a list of the peer review reports and author responses from that submission.
Round 1
Reviewer 1 Report
Comments and Suggestions for Authors
The paper could have some interest, although seems to be an spin off of a paper previously published in "Plants" and is somehow redundant. In the present form the paper presents many weak points and requires major improvement.
Introduction: the introduction is focused on the role of GSH in plants, which is very general, and only mentions at the end the role of peptides, and what is known on the peptide used in this study. Please rewrite the introduction focusiong on the role of peptides in plant growth and development, and what is known on the petide used in this study and how it was found.
Another major problem is that authors maintain that the petide activates the antioxidant activity by regulating the expression of SOD genes. This must be supported with the determination od SOD activity.
Methods: how is the peptide synthetized? How have the auhtors checked the stability? how it is applied. Please, provide all this information.
Figure 1: The figure showing the tolerance is not really convincing, as there are single plants. This tolerance is evidenced in young plants or is maintained at later phases of development? Provide pictures with more plants and clarify whether this tolerance is observed with older plants.
Figure 2, 3, 4, 5 and 5: please include the n number, and specify in the figure legend which statistical analysys have you used.
Figure 3: In the upper pannel letters are redundant with the lower pannels, pictures are out of focus, and some figures are cropped. No information can be obtained.
Tables 1 and 2: Please redo the tables and align the columns.
There are two figures 5. Please correct.
Taken altogether, the paper in the present form does not commit with the standards of quality of IJMS.
Author Response
Thank you very much for your professional review. Your comments and remarks were very helpful in revising the manuscript.
The paper could have some interest, although seems to be an spin off of a paper previously published in "Plants" and is somehow redundant. In the present form the paper presents many weak points and requires major improvement.
Introduction: the introduction is focused on the role of GSH in plants, which is very general, and only mentions at the end the role of peptides, and what is known on the peptide used in this study. Please rewrite the introduction focusiong on the role of peptides in plant growth and development, and what is known on the peptide used in this study and how it was found.
Reply: We agree with your remarks and have revised the Introduction section.
Another major problem is that authors maintain that the peptide activates the antioxidant activity by regulating the expression of SOD genes. This must be supported with the determination of SOD activity.
Reply: We agree that this conclusion would be desirable to confirm by determining SOD activity. Unfortunately, we could only determine FeSOD activity. Activity increased in the presence of AEDL, but only by about 11%. We also did not report data on FeSOD gene expression, since the change in gene activity was not convincing.
Methods: how is the peptide synthesized? How have the authors checked the stability? how it is applied. Please, provide all this information.
Reply: Sorry for our inaccuracy, we have added this data to the Materials and Methods section.
Figure 1: The figure showing the tolerance is not really convincing, as there are single plants. This tolerance is evidenced in young plants or is maintained at later phases of development? Provide pictures with more plants and clarify whether this tolerance is observed with older plants.
Reply: We grew tobacco for 28 days. Reference 40. We previously looked at tobacco tolerance for a longer period. It was noted that only in the presence of the AEDL peptide did tobacco tolerance to salt persist for 2-3 months. Morphometric parameters were previously determined on 30 samples in 3 replicates. These results are not presented in this manuscript.
Figure 2, 3, 4, 5 and 5: please include the n number, and specify in the figure legend which statistical analysis have you used.
Reply: Thank you for your comment, we have indicated the number n. The use of the ANOVA program is given in the Materials and Methods section.
Figure 3: In the upper panel letters are redundant with the lower panels, pictures are out of focus, and some figures are cropped. No information can be obtained.
Reply: Sorry, the figures were shifted as a result of adding text to the layout. We tried to fix it. The scale is indicated in Figure 3.
Tables 1 and 2: Please redo the tables and align the columns.
Reply: Sorry, tables were shifted as a result of adding text to the layout. We tried to fix it.
There are two figures 5. Please correct.
Reply: Sorry for the error, we have corrected the numbering of the figures

Reviewer 2 Report
Comments and Suggestions for Authors
The current paper devoted to analysis of effect of AEDL peptide on Nicotiana tabacum root growth and salt stress resistance.
Authors perform a large amount of experiments and analysing data.
However, there are a lot of confusion and text require major corrections.
The model require re-adjustement. It is not clear what did authors mean as stem cell niche. 3D image sand image analysis are required.
GSH is an regulator of auxin response and I would suggest to authors to test in situ auxin response with usage marker.
Details:
Fig.1, 3: scale bar!
Line 23: „Redox metabolism in plant cells inevitably includes the formation and accumulation of highly toxic reactive oxygen species (ROS) [1-3].” ¿? Redox metabolism can not induce ROS accumulation.
Line 24: “the level of ROS in plants increases. Conditions leading to damage to cellular organelles and cell membranes caused by ROS are called oxidative stress [4]” ¿?? This is not true. There is no level of ROS in plants because they are different in different cell type. This „conflict“ led to changes organ architecture and disturb development. Cell damage/death is cell specific and abundant only after high stress level.
Line 41: „The main function of this small molecule is due to its antioxidant“ ?? GSH have another function as antioxidant and have a relatively low capacity for ROS scavenger (to compare with ASC). https://doi.org/10.3390/biom10111550, https://doi.org/10.1016/j.plaphy.2013.10.028. https://doi.org/10.1007/s11627-009-9266-y
Lines 189: “ROS production was detected” ¿? It is ROS accumulation, not production. Moreover, you incubated root long time in water and wash with water, what in turn, changes ROS level. Since dye is not ratiometric, and require loading control. That’s why quantitative data is under a question. In addition, on Fig. 3 C, D – I did not understand what is RU? Relative to what?? Cell have different volume, different area and not be mechanistically compared. How you van build border between cell type?
For Nicotiana zonation you can consult: https://doi.org/10.1111/tpj.13631
Line 244: H2O2 is not a marker. Please, adjust layout and provide contents in µM/g. Moreover, plesae, specify cell types in shoot and root.
Line 301: what is normalization factor in fig. 3?
Table 3: layout.
Line 315: „GSH, mM/g“ ???? maybe µM?
Fig 4: normalization marker??
Figure 6: model is very speculative. No evidence and stem cell map in tobacco have been shown,
Lines 546 – 549: not clear how you cut seedlings. Can you provide any citation that in N.tabacum Cl 5 fold more important as P? Which reaction/process Cl invloved in your hand? Line 547 really claim this, but without citation.
Line 567: „from wheat roots“ ?????
Author Response
Thank you very much for your professional review. Your comments and remarks were very helpful in revising the manuscript.
The current paper devoted to analysis of effect of AEDL peptide on Nicotiana tabacum root growth and salt stress resistance.
Authors perform a large amount of experiments and analysing data.
However, there are a lot of confusion and text require major corrections.
The model require re-adjustement. It is not clear what did authors mean as stem cell niche. 3D image sand image analysis are required.
Reply: We agree that the model requires improvement, as any model does. We plan to continue research in this direction. By the term stem cell niche we mean the center of resting cells. We do not provide a 3-D image in this manuscript.
GSH is an regulator of auxin response and I would suggest to authors to test in situ auxin response with usage marker.
Reply: Thank you for a very interesting proposal. Unfortunately. at the moment there are certain difficulties, we had such plans.
Details:
Fig.1, 3: scale bar!
Reply: Sorry, we have inserted scale bars in figures 1 and 3.
Line 23: „Redox metabolism in plant cells inevitably includes the formation and accumulation of highly toxic reactive oxygen species (ROS) [1-3].” ¿? Redox metabolism can not induce ROS accumulation.
Reply: In this sentence we used the word ROS level, meaning the total amount of ROS. However, we agree with you and have rewritten the entire phrase.
We have made corrections to the manuscript text
Line 24: “the level of ROS in plants increases. Conditions leading to damage to cellular organelles and cell membranes caused by ROS are called oxidative stress [4]” ¿?? This is not true. There is no level of ROS in plants because they are different in different cell type. This „conflict“ led to changes organ architecture and disturb development. Cell damage/death is cell specific and abundant only after high stress level.
Reply: We agree with you. Each cell type has its own ROS signature. However, cell damage and death can occur not only after high levels of stress and prolonged exposure, but also in very sensitive plant species and varieties.
Line 41: „The main function of this small molecule is due to its antioxidant“ ?? GSH have another function as antioxidant and have a relatively low capacity for ROS scavenger (to compare with ASC). https://doi.org/10.3390/biom10111550, https://doi.org/10.1016/j.plaphy.2013.10.028. https://doi.org/10.1007/s11627-009-9266-y
Reply: Thank you for the references. We used one of your references and added it to the list of references.
We are very sorry that our communication with Klaus Palme was interrupted. You have excellent equipment and can do work at a very high level.
We agree with you, glutathione has other functions. Other functions of glutathione were mentioned below. We have combined information about the most important functions of glutathione in one place.
Lines 189: “ROS production was detected” ¿? It is ROS accumulation, not production. Moreover, you incubated root long time in water and wash with water, what in turn, changes ROS level. Since dye is not ratiometric, and require loading control. That’s why quantitative data is under a question. In addition, on Fig. 3 C, D – I did not understand what is RU? Relative to what?? Cell have different volume, different area and not be mechanistically compared. How you van build border between cell type?
Reply: We did not incubate the roots in water, we grew the roots in liquid M-C medium and before detection we washed the roots 3 times with M-C medium without additives. Our task was to detect ROS in comparison with the control root. The root zones are clearly visually distinct and visible under a microscope. This is the accepted zoning in the scientific literature. The software (specified in Materials and Methods) was used to measure the fluorescence intensity.
For Nicotiana zonation you can consult: https://doi.org/10.1111/tpj.13631
Reply: Thanks for the reference.
Line 244: H2O2 is not a marker. Please, adjust layout and provide contents in µM/g. Moreover, plesae, specify cell types in shoot and root.
Reply: H2O2 is one of the most important and widespread ROS and is an important signaling molecule in abiotic stress. Therefore, H2O2 can be a marker of abiotic stress. Thank you for the comment, we have corrected the H2O2 content in μM. The H2O2 content was determined separately in shoots and roots, without separating them by cell types. Unfortunately, we did not have the ability to separate them by cell types.
Line 301: what is normalization factor in fig. 3?
Reply: If we mean the normalization factor in table 3, then it is 0.01 M 5,5’-Dithio-Bis-(2-Nitrobenzoic Acid) alcohol solution
Table 3: layout.
Reply: Sorry, there was a technical error.
Line 315: „GSH, mM/g“ ???? maybe µM?
Reply: Sorry, of course in mkM
Fig 4: normalization marker??
Reply: GaPDh (LOC107828122) was used as a reference gene.
Figure 6: model is very speculative. No evidence and stem cell map in tobacco have been shown,
Reply: We agree that the presented scheme is a model. It was proposed based on the literature, our data obtained earlier and the data presented in this paper. It is a basic model that provides a direction for further research.
Lines 546 – 549: not clear how you cut seedlings. Can you provide any citation that in N.tabacum Cl 5 fold more important as P? Which reaction/process Cl invloved in your hand? Line 547 really claim this, but without citation.
Reply: Sorry, there was a mistake, we have improved the presented methodology and provided a reference
Line 567: „from wheat roots“ ?????
Reply: Sorry, there was a mistake, of course these are tobacco roots

Round 2
Reviewer 1 Report
Comments and Suggestions for Authors
After carefully reading the revised version, I can say that the introduction has been dramatically improved, but the problems with the experiments and the results are still there.
Some essential data has been published elsewhere, but the authors need to cite where in the answer letter. The phenotype is not convincing in the presented picture, and also the model is not supported by the SOD determinations. I'm afraid I have to disagree with the authors that they can only determine one of the activities, as there are plenty of protocols in the literature to determine such activity.
So, I cannot recommend this paper for publication.
Reviewer 2 Report
Comments and Suggestions for Authors
Thank you, some points have been answered, but some points remained unclear and discussion is very confusing and speculative. No direct evidence of the model have been provided.
Plesae, make modell less speculative!
Figure 3: description is not clear.
Line 441: "superoxide ion can accumulate in the RGF1-PLT signaling pathway, which can lead 441 to oxidation" ???? Superoxide can not accumuate in pathway.
Line 488: "GP expression levels were so low that they were not discussed" -this istrue because GP does not have a significant role in H2O2 scavenging. The main role is APX.
Lines 512- 521: citation 61 is baout shoot stem cell niche what is different form root: (from in shoot and to in roots). Moreover, what did authors mean as redox balance? H2O2, or O2- accumulation? Or GSH/GSSG ratio or ASC/DHA ratio? or else? There is no ROS accumulation itself without specification of cell/compartments.
Figure 7 (model) need to be based on results, not on speculations. Where is WUS (may be WOX5-like??) in Nicotiana? What is stem cell differentiation? Ascorbta is a mainplayer in stem cell niche maitenamce. But there is no ASC mention/measure in the paper.
Line 553: There is still no answer why Nicotiana plants (control) reuire almost 5 fold more Cl as P. 6 mM Cl and only 1,25 mM P. Perhaps, you may now reason. https://www.duchefa-biochemie.com/product/details/number/M0222
Line 561: "our method [40]"?? More details need. Any washing/transfer to new medium can significantly affected results.
Line 573: "Absorption Absorbance" ??